# Chitosan Spraying Enhances the Growth, Photosynthesis, and Resistance of Continuous *Pinellia ternata* and Promotes Its Yield and Quality

**DOI:** 10.3390/molecules28052053

**Published:** 2023-02-22

**Authors:** Fengfeng Chen, Qinju Li, Yue Su, Yang Lei, Cheng Zhang

**Affiliations:** 1School of Public Health, Guizhou Medical University, Guiyang 550025, China; 2Department of Food and Medicine, Guizhou Vocational College of Agriculture, Qingzhen 551400, China

**Keywords:** continuous cropping obstacle, alleviation, growth, resistance, quality

## Abstract

The continuous cropping obstacle has become the key factor that seriously restricts the growth, yield, and quality of *Pinellia ternata*. In this study, the effects of chitosan on the growth, photosynthesis, resistance, yield, and quality of the continuous cropping of *P. ternata* were investigated by two field spraying methods. The results indicate that continuous cropping significantly (*p* < 0.05) raised the inverted seedling rate of *P. ternata* and inhibited its growth, yield, and quality. Spraying of 0.5~1.0% chitosan effectively increased the leaf area and plant height of continuous *P. ternata*, and reduced its inverted seedling rate. Meanwhile, 0.5~1.0% chitosan spraying could notably increase its photosynthetic rate (Pn), intercellular carbon dioxide concentration (Ci), stomatal conductance (Gs), and transpiration rate (Tr), and decrease its soluble sugar, proline (Pro), and malonaldehyde (MDA) contents, as well as promoting its superoxide dismutase (SOD), peroxidase (POD), and catalase (CAT) activities. Additionally, 0.5~1.0% chitosan spraying could also effectively enhance its yield and quality. This finding highlights that chitosan can be proposed as an alternative and practicable mitigator for alleviating the continuous cropping obstacle of *P. ternata*.

## 1. Introduction

*Pinellia ternata* (Thunb.) Breit., a medical and ornamental perennial herb of *Pinellia* in the *Araceae* family, has been widely planted in northwest and southwest China [1]. It is rich in alkaloids, proteins, polysaccharides, flavonoids, nucleosides, phenolics, minerals, and so on [2,3,4]. As an important traditional Chinese medicine, its dried bulbs’ properties are moderate, and it is widely used for stopping vomit, breaking up sputum, eliminating swelling, dispersing knots, reducing hematic fat, protecting the liver, treating coronary disease, and for its antitumor properties [5,6,7,8]. Recently, the *P. ternata* industry has flourished rapidly in Guizhou and Gansu Provinces in China, with cropping areas of over 4000 and 1333 hm^2^, respectively, and made an energetic contribution to alleviating poverty and revitalizing rural areas [1]. Because of the important medicinal value and industrial prospects of *P. ternata*, the cultivation measures to promote its growth, yield, and medicinal quality have attracted increasing attention.

Unfortunately, after continuous cropping of *P. ternata*, its growth and development are inhibited, its quality deteriorates, its susceptibility to diseases and pests is intensified, and its yield declines [9,10,11,12]. Meanwhile, the soil in which it is planted becomes acidified, the soil nutrients are sealed, the microbial community structure in the soils becomes unbalanced, and soils need to be replanted at an interval of more than 7 to 8 years [12,13,14]. Additionally, farmers often maintain the yield of *P. ternata* by increasing fertilizer use and abusing pesticides, which also leads to problems such as agricultural product quality safety and environmental pollution. Thus, the continuous cropping obstacle has become the key factor that seriously restricts the growth, quality, yield and development of *P. ternata*, and the establishment of its prevention and control strategy has become a hot issue in research. Currently, soil disinfection, intercropping, and rotation measures are often used to alleviate the continuous cropping obstacle of *P. ternata* [13,14,15]. For example, He et al. [13] reported that crop rotation could remediate the deteriorated microbial structure in continuous cropping soils of *P. ternata*. Hang et al. [15] found that intercropping soybean could increase the yield of *P. ternata*, and intercropping pepper could enhance the succinic acid and guanosine contents of *P. ternata*. However, the application and promotion of these measures are limited by factors including operability, low efficiency, and environmental pollution, etc. Considering the serious harmfulness of the continuous cropping obstacle, more alternative and practicable control measures need to be developed for enhancing the healthy development of the *P. ternata* industry.

Chitosan, a natural, nontoxic to humans or other organisms, *N*-deacetylation product of chitin, has been widely applied in food, agriculture, medicine, and cosmetic fields due to its advantages including nontoxicity, bioactivity, biodegradability, and renewability [16,17,18,19]. As a promising natural resource in agriculture, it has often been applied as a plant growth promoter, biological fertilizer, resistance inducer, or biological fungicide [19,20,21,22,23,24]. It can stimulate various physiological processes in plants including cell division and elongation, nutrient uptake, stress resistance, etc., thereby enhancing their growth, yield, and quality [19,25,26,27,28]. For instance, Li et al. [27,29] reported that chitosan, and its combination with allicin, could effectively enhance *Rosa roxburghii* resistance, growth, yield, and quality. Pan et al. [30] indicated that chitosan could improve the radicle and germ length, plant height, fresh weight, and dry weight of *Trifolium repens* under salt stress. Li et al. [31] reported that chitosan could promote the drought resistance of *Sctellaria baicalensis* seedlings. Recently, Liu et al. [32] indicated that chitosan soaking enhanced the growth, photosynthesis, resistance, yield, and quality of *Platycodon grandifloras*. So far, there has been no reports in the literature of whether chitosan promotes the growth of *P. ternata*. Accordingly, whether chitosan can enhance the growth, resistance, yield, and quality of continuous *P. ternata*, and alleviate its continuous cropping obstacle, is worth further study.

In this work, the influence of chitosan on the growth (leaf area, plant height, stem diameter, and inverted seedling rate) and photosynthesis (chlorophyll, photosynthetic rate (Pn), intercellular carbon dioxide concentration (Ci), stomatal conductance (Gs), and transpiration rate (Tr)) of the continuous *P. ternata* was evaluated. Simultaneously, the stress resistance capacity of plants strongly affects their growth; soluble sugar, proline (Pro), malonaldehyde (MDA), superoxide dismutase (SOD), peroxidase (POD), and catalase (CAT) are all closely connected to stress resistance. Thus, the influences of chitosan on the abovementioned resistance parameters of continuous *P. ternata* was assessed. Finally, the influences of chitosan spraying on its yield and quality were also investigated. The aim of this study is to provide a new and alternative strategy for alleviating the continuous cropping obstacle of *P. ternata*.

## 2. Results

### 2.1. Influences of Chitosan on the Growth of P. ternata

The influence of chitosan spraying on the leaf area, plant height, and stem diameter of *P. ternata* plants is shown in Table 1. The leaf area of continuous *P. ternata* for 2 years, at the vigorous growth and inverted seedling periods, was significantly (*p* < 0.05) inferior to that of non-continuous *P. ternata*; and the plant height at the full seedling, vigorous growth, and inverted seedling periods was also significantly (*p* < 0.05) inferior to that of non-continuous *P. ternata*; the stem diameter at the three growth periods was also lower than that of non-continuous *P. ternata*. Compared with plants not sprayed with chitosan, 0.5~1.0% chitosan spraying significantly (*p* < 0.05) enhanced the leaf area of continuous *P. ternata* plants at the vigorous growth and inverted seedling periods, and the plant height of continuous *P. ternata* plants at the full seedling, vigorous growth, and inverted seedling periods. Additionally, compared with non-continuous *P. ternata*, the leaf area, plant height, and stem diameter of continuous *P. ternata* plants treated by 0.1~1.0% chitosan spraying were still inferior. The present results indicate that the continuous cropping significantly (*p* < 0.05) decreased the leaf area, plant height, and stem diameter of *P. ternata* plants, and that 0.5~1.0% chitosan spraying had a promoting effect on the growth of continuous *P. ternata* plants.

The inverted seedling is a physiological phenomenon that often occurs during the growth of *P. ternata*, which is a dormancy to resist the adverse environment. During the growth of *P. ternata*, when the environmental conditions such as temperature, humidity, light, allelochemicals, etc. change greatly, its aboveground stems and leaves will gradually wither, and it makes underground stems (inverted seedling) to survive the adverse environment. The influence of chitosan on the inverted seedling rate of *P. ternata* plants is displayed in Figure 1. Continuous cropping significantly (*p* < 0.05) increased the inverted seedling rate of *P. ternata* plants, 0.5~1.0% chitosan spraying effectively (*p* < 0.05) decreased the inverted seedling rate of the continuous *P. ternata* plants. However, the inverted seedling rate in plants treated by 0.1~1.0% chitosan significantly (*p* < 0.05) exceeded that of non-continuous *P. ternata* plants. These findings emphasize that 0.5~1.0% chitosan spraying could effectively decrease the inverted seedling rate of *P. ternata*, and alleviate its cropping obstacle in part, but not completely, which might be related to chitosan application enhancing the stress resistance of *P. ternata*.

### 2.2. Influences of Chitosan on Photosynthetic Capacity of P. ternata

As shown in Figure 2, the chlorophyll content of the continuous *P. ternata* leaves treated by 0.1~1.0% chitosan spraying was lower than that of non-continuous *P. ternata* leaves, and the continuous cropping or chitosan treatment had no significant influence on the *P. ternata* chlorophyll content. The influences of chitosan spraying on the Pn, Ci, Gs, and Tr contents of *P. ternata* are shown in Figure 3. The Pn, Ci, Gs, and Tr of the continuous *P. ternata* were significantly (*p* < 0.05) lower than those of the non-continuous *P. ternata*. Compared with plants not sprayed with chitosan, 0.5~1.0% chitosan spraying significantly (*p* < 0.05) enhanced the Pn and Ci contents of the continuous *P. ternata*, and 0.1~1.0% chitosan spraying significantly (*p* < 0.05) enhanced the Gs and Tr contents. Meanwhile, the Pn, Ci, Gs, and Tr contents of the continuous *P. ternata* treated by 0.1~1.0% chitosan spraying were significantly (*p* < 0.05) smaller than those of the non-continuous *P. ternata*. The results presented here indicate that the foliar application of 0.5~1.0% chitosan spraying effectively increased the Pn, Ci, Gs, and Tr contents of the continuous *P. ternata*, thereby promoting favorable growth.

### 2.3. Influence of Chitosan on Stress Resistance of P. ternata

The influence of chitosan spraying on the soluble sugar, Pro, MDA, SOD activity, POD activity, and CAT activity of *P. ternata* plants is displayed in Figure 4. The continuous cropping significantly (*p* < 0.05) raised the soluble sugar, Pro, and MDA contents of *P. ternata* compared with non-continuous cropping, and decreased the SOD, POD, and CAT activities. Compared with non-sprayed chitosan, 0.1~1.0% chitosan spraying significantly (*p* < 0.05) decreased the soluble sugar, Pro, and MDA contents of the continuous *P. ternata*, and 0.5~1.0% chitosan spraying significantly (*p* < 0.05) enhanced its SOD, POD, and CAT activities. Meanwhile, the soluble sugar, Pro, and MDA contents of the continuous *P. ternata* treated by 0.1~1.0% chitosan spraying were significantly (*p* < 0.05) higher than those of non-continuous *P. ternata*, and its SOD, POD, and CAT activities were significantly (*p* < 0.05) inferior to those of non-continuous *P. ternata*. These results show that continuous cropping significantly (*p* < 0.05) decreased the stress resistance of *P. ternata* plants, and 0.5~1.0% chitosan spraying could effectively decrease the soluble sugar, Pro, and MDA contents of continuous *P. ternata* and promote its SOD, POD, and CAT activities, thereby enhancing the stress resistance of continuous *P. ternata*.

### 2.4. Influences of Chitosan on Yield and Quality of P. ternata

As shown in Table 2, the continuous cropping significantly (*p* < 0.05) decreased the seed weight, medicinal material weight, and total yield of *P. ternata*, compared with the non-continuous cropping plants. Spraying with 0.1~1.0% chitosan led to an increase in the yield for the continuous *P. ternata*, with the mean seed weight, medicinal material weight, and total yield of 27.56–46.86, 85.69–158.52, and 113.25–205.38 kg per 667 m^2^, respectively, significantly (*p* < 0.05) higher ((1.17–1.99)-fold, (1.09–2.01)-fold, (1.10–2.00)-fold, respectively) than for the non-sprayed plants. Meanwhile, the seed weight, medicinal material weight, and total yield of the continuous *P. ternata* treated by 0.1~1.0% chitosan spray were still significantly (*p* < 0.05) lower than those of the non-continuous *P. ternata*. The results indicate that 0.1~1.0% chitosan spraying could effectively alleviate the cropping obstacle of *P. ternata* and enhance its bulb growth and yield formation.

The influence of chitosan spraying on the water, ash, extractum, and succinic acid of *P. ternata* is shown in Figure 5. Continuous cropping significantly (*p* < 0.05) reduced the water, ash, extractum, and succinic acid contents of *P. ternata* compared with non-continuous cropping plants. Compared with plants not sprayed with chitosan, 0.5~1.0% chitosan spraying significantly (*p* < 0.05) increased the water, ash, extractum, and succinic acid contents of continuous *P. ternata*. However, the water, ash, extractum, and succinic acid contents of continuous *P. ternata* treated by 0.5~1.0% chitosan spraying were still significantly (*p* < 0.05) lower than those of non-continuous *P. ternata*. These findings indicate that 0.5~1.0% chitosan spraying effectively improved the medicinal quality of continuous *P. ternata* bulbs.

## 3. Discussion

The continuous cropping obstacle seriously restricts the growth, quality, yield and development of *P. ternata* [9,10,11,12,13,14]. Xiao [33] found that continuous cropping obviously inhibited the growth, yield, and quality of *P. ternata*, and continuous cropping for 2 years exhibited a stronger inhibitory effect compared with continuous cropping for 1, 3, 4, and 5 years. In this study, *P. ternata* in the continuous cropping group for 2 years was used as the treatment object, and *P. ternata* in the non-continuous cropping group was used as the intergroup control. The results indicate that continuous cropping for 2 years significantly (*p* < 0.05) increased the inverted seedling rate of *P. ternata* and inhibited its growth, yield, and quality, and caused it physiological damage and put it in a state of stress, which was consistent with previous results. Chitosan can promote cell division and elongation, via activating the gene expression and signal transduction of cytokinin and auxin, which enhance the nutrient intake and growth of plants [19,23,24,25]. Pan et al. [30] found that chitosan could improve the radicle length, germ length, and plant height of *Trifolium repens* under salt stress. Liu et al. [32] reported that chitosan soaking could also enhance the cotyl length, radicle length, leaf area, plant height, and stem diameter of *Platycodon grandifloras*. In this work, 0.5~1.0% chitosan spraying significantly (*p* < 0.05) increased the leaf area and plant height of continuous *P. ternata*, and decreased its inverted seedling rate. These results emphasize that 0.5~1.0% chitosan spraying could effectively alleviate the cropping obstacle of *P. ternata*.

Photosynthesis is the physiological basis of plant life action, while chitosan is an enhancer of photosynthesis [26]. Liu et al. [32] demonstrated that chitosan soaking could also enhance *Platycodon grandifloras* photosynthesis. Our previous research also showed that chitosan spraying could promote the photosynthesis of *Actinidia chinensis* and *Rosa roxburghii* [34,35]. In this study, although 0.1~1.0% chitosan spraying could not significantly (*p* < 0.05) increase the chlorophyll content of the continuous *P. ternata*, 0.5~1.0% chitosan spraying could effectively enhance its Pn, Ci, Gs, and Tr contents, which further highlights that chitosan spraying could effectively alleviate the cropping obstacle of *P. ternata*. The osmotic potential of plant tissue under stress can be regulated by soluble sugar, to maintain plant growth, and Pro can effectively protect the structural integrity of cell films, while MDA is an indicator of membrane damage and stress resistance degrees [36]. SOD and CAT can scavenge plants’ free radicals, and POD can catalyze H_2_O_2_ decomposition in lignin biosynthesis, as such they are connected to stress resistance [36]. Li et al. [31] found that chitosan could promote the drought resistance of *Sctellaria baicalensis* seedlings, and Dzung et al. [18] indicated that chitosan improved the drought resistance of *Coffea* spp., as well as the disease resistance of *Actinidia chinensis* and *Rosa roxburghii* [34,37,38]. In this work, continuous cropping significantly (*p* < 0.05) increased the soluble sugar, Pro, and MDA contents of *P. ternata*, and decreased its SOD, POD, and CAT activities, as well as put it in a state of stress. Effectively, 0.5~1.0% chitosan spraying could significantly (*p* < 0.05) decrease the soluble sugar, Pro, and MDA contents of the continuous *P. ternata*, and promote its SOD, POD, and CAT activities, thereby enhancing the stress resistance of the continuous *P. ternata*. These results also demonstrate that 0.5~1.0% chitosan has a notably alleviatory effect on the cropping obstacle of *P. ternata*.

Good growth is the basis for the high yield and superior quality of *P. ternata*. Pan et al. [29] indicated that chitosan could increase the fresh weight and dry weight of *Trifolium repens* under salt stress. Liu et al. [32] reported that chitosan soaking could enhance the yield and quality of *Platycodon grandifloras*, and Li et al. [27,29] also indicated that chitosan could effectively improve the yield and quality of *Rosa roxburghii*. In this study, continuous *P. ternata* sprayed with 0.1~1.0% chitosan exhibited an increased yield, with the seed weight, medicinal material weight, and total yield. Spraying with 0.5~1.0% chitosan effectively enhanced the water, ash, extractum, and succinic acid of continuous *P. ternata* bulbs. These results emphasize that chitosan is an effective mitigator for alleviating the continuous cropping obstacle of *P. ternata.* However, the growth, photosynthesis, resistance, yield, and quality of the continuous *P. ternata* treated by 0.5~1.0% chitosan spraying were still significantly (*p* < 0.05) inferior to those of non-continuous *P. ternata*. This finding highlights that 0.5~1.0% chitosan spraying could effectively alleviate the cropping obstacle of *P. ternata*, but cannot completely remove it.

Overall, the possible mechanism of how chitosan mediated the alleviation of the continuous cropping obstacle of *P. ternata* is as follows: chitosan might be used as a plant growth promoter or biological fertilizer to enhance the growth and photosynthesis of continuous *P. ternata*, and might also be used as an inducer to improve its stress resistance and reduce its inverted seedling rate, as well as might also be used as a biological fungicide to prevent disease occurrence. These possible positive effects together, enhanced the growth, yield, and quality of continuous *P. ternata* and alleviated its continuous cropping obstacle. In the future, the molecular biological mechanism of the chitosan-mediated alleviation of the continuous cropping obstacle of *P. ternata* needs to be studied. Meanwhile, chitosan is a natural biopolymer, with nontoxic, renewable, and biodegradable advantages [19,20,37], and the safe interval period, of 133 days, for *P. ternata* was very long. Accordingly, the safety risk caused by chitosan was almost nonexistent. This work emphasizes that chitosan can be proposed as a practicable promotor or mitigator for promoting the growth, photosynthesis, resistance, yield, and quality of continuous *P. ternata*, and alleviating its continuous cropping obstacle.

## 4. Materials and Methods

### 4.1. Chemicals and Seeds

Water-soluble chitosan (deacetylation ≥ 90.00%) was produced by Mingrui Bioengineering Co., Ltd. (Zhenzhou, China). Seed bulbs of ball *P. ternata* (diameter of 0.6~1.2 cm) were provided by the Institute of Modern Chinese Herbal Medicines, Guizhou Academy of Agricultural Sciences (Guiyang, China). Other chemicals were analytical or chromatographic grade.

### 4.2. P. ternata Herb Garden

The *P. ternata* herb garden was divided into two parts: One part had never been cultivated with *P. ternata* (i.e., no continuous cropping), and the other part had been cultivated with *P. ternata* in the past two years. In the third year, the chitosan spraying experiment was carried out, and the experimental groups included the no continuous cropping group and the continuous cropping for 2 years group. *P. ternata* was cultivated by ridging, and each treated plot area was 3.6 m^2^ (ridge width 1.2 m, length 3 m, ridge between 0.2 m). Seed bulbs of ball *P. ternata* were broadcast-sown on 15 March 2022, then covered with 5 cm of soil, and the application dosage of the seed bulbs was 140 kg per 667 m^2^. The basic information of the *P. ternata* herb garden is shown in Table 3.

### 4.3. Chitosan Spraying Experiment

The foliar spray method and completely randomized method were used for spraying chitosan and delineating experimental plots, respectively. Five treatments were designed: (1) continuous *P. ternata* treated by clear water (Ch 0.0, intragroup control), (2) continuous *P. ternata* treated by 0.1% chitosan dilution liquid (Ch 0.1), (3) continuous *P. ternata* treated by 0.5% chitosan dilution liquid (Ch 0.5), (4) continuous *P. ternata* treated by 1.0% chitosan dilution liquid (Ch 1.0), and (5) non-continuous *P. ternata* treated by clear water (NCC, intergroup control). Each treatment had three replicates, with a total of 15 plots. The *P. ternata* plants were sprayed by chitosan dilution liquid using an electrostatic atomizer (Qiming Machinery Co., Ltd., Taizhou, Zhejiang, China) on April 15 and April 30. The application dosage of the chitosan dilution liquid was 60 L per 667 m^2^.

### 4.4. Analytical Method

#### 4.4.1. Growth Parameters

The leaf area, height, and stem diameter of the *P. ternata* plants were monitored at full seedling period (April 30), vigorous growth period (May 15), and inverted seedling period (May 30). The stem diameter was measured by a vernier caliper, and plant height, and leaf length and width, were measured by a ruler. The leaf area coefficient method was used to calculate the main leaf area of *P. ternata*:S = k·a·b(1)
where S is the main leaf area, k is a coefficient of 0.666, a is the leaf length, and b is the leaf width.

#### 4.4.2. Photosynthesis Parameters

Fifteen *P. ternata* plants, on the east, west, south, north, and in the middle of each plot, were randomly selected at the vigorous growth period (May 15) for investigation of their photosynthesis parameters. The chlorophyll content of the fully-expanded leaves of *P. ternata* was monitored by a SPAD-502Plus chlorophyll meter (Konica Minolta Inc., Osaka, Japan). Meanwhile, their Pn, Ci, Gs, and Tr contents were monitored by a portable LI-6400XT photosynthesis measurement system (LI-COR Inc., Lincoln, NE, USA), with the photosynthetically active radiation of 1000 μmol m^−2^ s^−1^ at 8:00–10:00 a.m. [27,28].

#### 4.4.3. Resistance Parameters

The fresh plants with the same orientation were collected and taken back to the laboratory to determine the resistance indexes of *P. ternata*, such as soluble sugar, Pro, MDA, SOD activity, POD activity, and CAT activity, according to the methods of Wang et al. [38] and Zhang et al. [39,40]. Anthrone colorimetric, ninhydrin colorimetric, and thiobarbituric acid methods were used for determining the soluble sugar, Pro, and MDA contents, respectively. The nitrogen blue tetrazole, guaiacol, and potassium permanganate titration methods were used to determine SOD, POD, and CAT activities, respectively.

#### 4.4.4. Yield and Quality Parameters

The underground bulbs of *P. ternata* in each plot were collected and washed on September 10. Among them, those with a diameter greater than 1.2 cm were selected as medicinal materials, and those with a diameter less than 1.2 cm were reserved as seeds. Their weights were determined by the gravimetric method. The water, ash, extractum, and succinic acid of the *P. ternata* bulbs were determined as described by the general principles of four parts of *Chinese Pharmacopoeia*, 2020 [8].

### 4.5. Statistical Analyses

The data represent mean values ± standard deviation (SD) of three replicates. The Duncan’s test, with one-way analysis of variance (ANOVA), on the SPSS 18.0 software (SPSS Inc., Chicago, IL, USA), was used for analyzing significant differences. The Origin 10.0 software (OriginLab, Northampton, MA, USA) was used to create the figures.

## 5. Conclusions

In conclusion, 0.5~1.0% chitosan spraying could effectively improve the leaf area and plant height of continuous *P. ternata* and reduce its inverted seedling rate. Meanwhile, 0.5~1.0% chitosan spraying notably enhanced the photosynthesis and resistance of continuous *P. ternata*, and effectively improved its yield and quality. This study emphasizes that chitosan can be proposed as an enhancer or mitigator for promoting the growth, yield, and quality of continuous *P. ternata*, and alleviating its continuous cropping obstacle. However, 0.5~1.0% chitosan spraying could not completely alleviate the cropping obstacle of *P. ternata*. Thus, the molecular biological mechanism of chitosan for alleviating the continuous cropping obstacle of *P. ternata*, and the screening of its synergistic surfactant or adjuvant, need to be studied in the future.

## Figures and Tables

**Figure 1 molecules-28-02053-f001:**
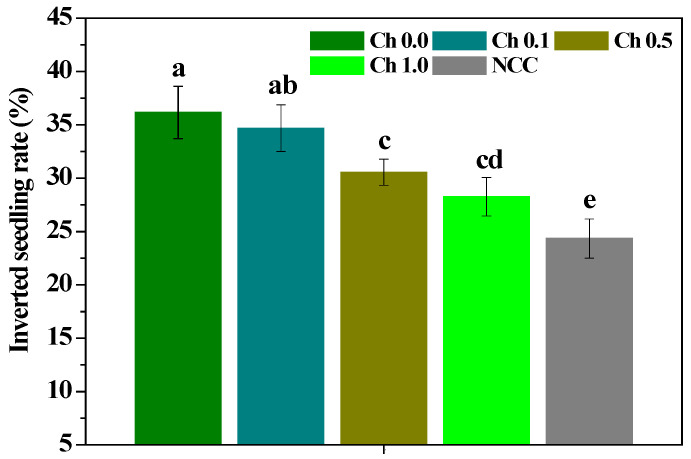
The influence of chitosan spraying on the inverted seedling rate of *P. ternata*. Error bars show SD, and different letters indicate significant differences at 5% (*p* < 0.05) level.

**Figure 2 molecules-28-02053-f002:**
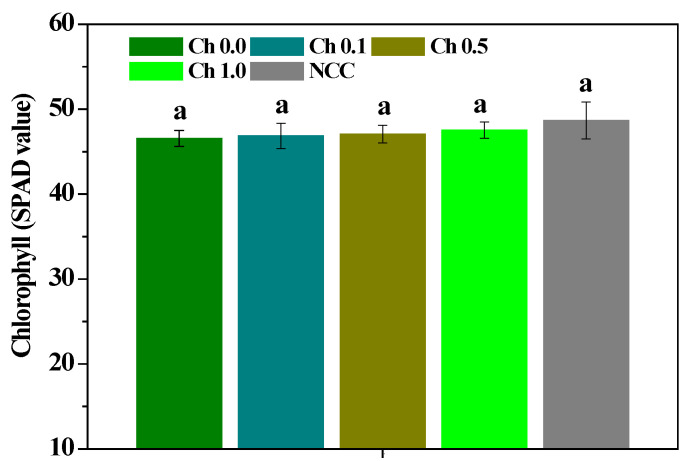
The influence of chitosan spraying on the chlorophyll content of *P. ternata* leaves. Error bars show SD, and different letters indicate significant differences at 5% (*p* < 0.05) level.

**Figure 3 molecules-28-02053-f003:**
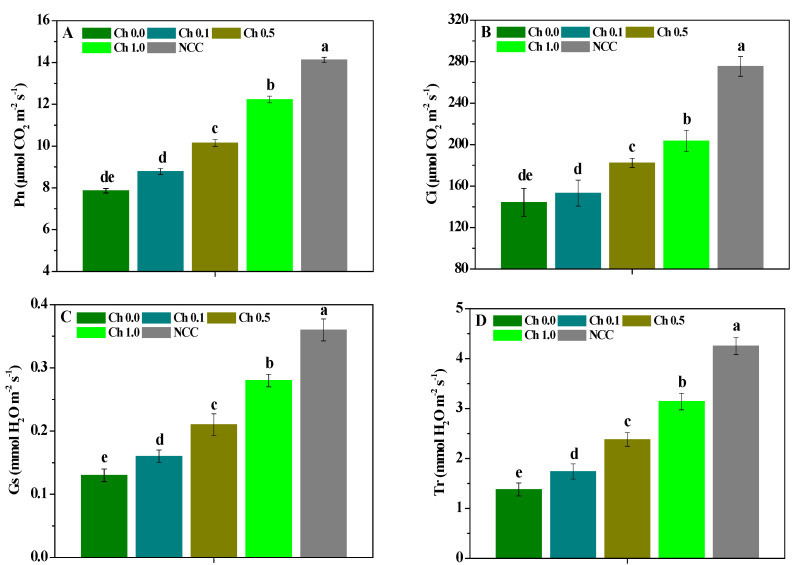
The influence of chitosan spraying on the Pn (**A**), Ci (**B**), Gs (**C**), and Tr (**D**) contents of *P. ternata* leaves. Error bars show SD, and different letters indicate significant differences at 5% (*p* < 0.05) level.

**Figure 4 molecules-28-02053-f004:**
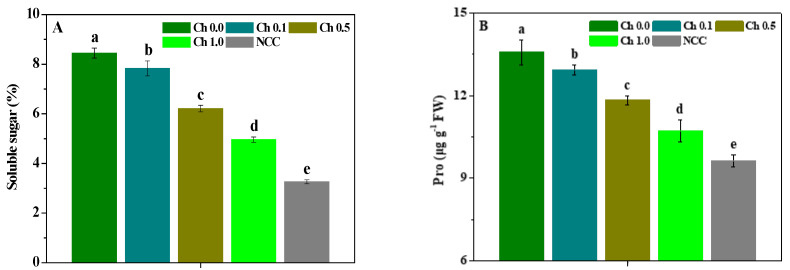
The influence of chitosan spraying on the soluble sugar (**A**), Pro (**B**), MDA (**C**), SOD activity (**D**), POD activity (**E**), and CAT activity (**F**) of *P. ternata*. Error bars show SD, and different letters indicate significant differences at 5% (*p* < 0.05) level.

**Figure 5 molecules-28-02053-f005:**
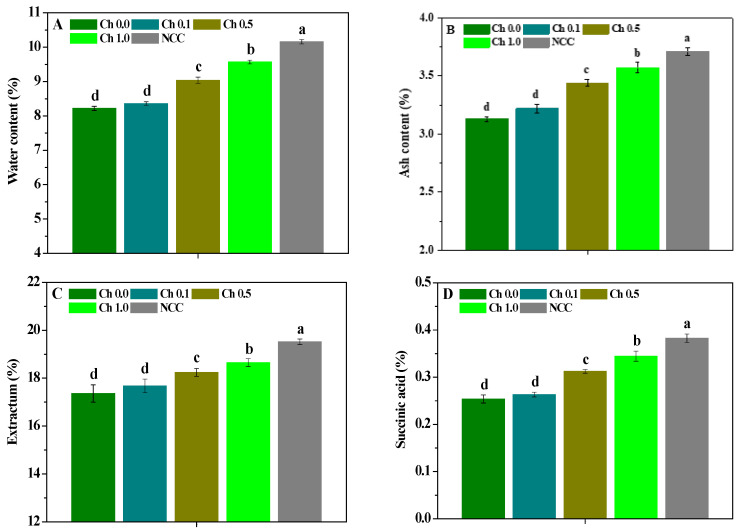
The influence of chitosan spraying on the water (**A**), ash (**B**), extractum (**C**), and succinic acid (**D**) of *P. ternata*. Error bars show SD, and different letters indicate significant differences at 5% (*p* < 0.05) level.

**Table 1 molecules-28-02053-t001:** The influences of chitosan spraying on the leaf area, height, and stem diameter of *P. ternata* plants.

Growing Periods	Treatments	Leaf Area (cm^2^)	Plant Height (cm)	Stem Diameter (mm)
Full seedling period	Ch 0.0	8.28 ± 0.07 ^a^	4.08 ± 0.03 ^d^	2.08 ± 0.05 ^a^
	Ch 0.1	8.37 ± 0.08 ^a^	4.16 ± 0.04 ^d^	2.11 ± 0.06 ^a^
	Ch 0.5	8.24 ± 0.06 ^a^	4.86 ± 0.06 ^c^	2.12 ± 0.03 ^a^
	Ch 1.0	8.31 ± 0.11 ^a^	5.15 ± 0.09 ^b^	2.09 ± 0.05 ^a^
	NCC	8.35 ± 0.07 ^a^	5.93 ± 0.05 ^a^	2.14 ± 0.05 ^a^
Vigorous growth period	Ch 0.0	10.86 ± 0.36 ^cd^	6.12 ± 0.24 ^c^	2.09 ± 0.06 ^a^
	Ch 0.1	11.24 ± 0.06 ^c^	6.57 ± 0.25 ^bc^	2.12 ± 0.05 ^a^
	Ch 0.5	12.88 ± 0.18 ^c^	6.94 ± 0.09 ^b^	2.15 ± 0.07 ^a^
	Ch 1.0	13.36 ± 0.19 ^b^	7.13 ± 0.18 ^ab^	2.10 ± 0.03 ^a^
	NCC	15.06 ± 0.15 ^a^	7.68 ± 0.14 ^a^	2.16 ± 0.05 ^a^
Inverted seedling period	Ch 0.0	12.92 ± 0.22 ^d^	6.72 ± 0.16 ^d^	2.13 ± 0.06 ^a^
	Ch 0.1	13.16 ± 0.28 ^cd^	6.89 ± 0.07 ^cd^	2.16 ± 0.07 ^a^
	Ch 0.5	13.75 ± 0.24 ^bc^	7.18 ± 0.15 ^c^	2.18 ± 0.06 ^a^
	Ch 1.0	14.09 ± 0.17 ^b^	7.51 ± 0.10 ^b^	2.21 ± 0.04 ^a^
	NCC	15.13 ± 0.20 ^a^	8.04 ± 0.11 ^a^	2.24 ± 0.05 ^a^

Values show the mean ± SD, and different letters indicate significant differences at 5% (*p* < 0.05) level.

**Table 2 molecules-28-02053-t002:** The influence of chitosan spraying on the seed weight, medicinal material weight, and total yield of *P. ternata*.

Treatments	Seed Weight (kg per 667 m^2^)	Medicinal Material Weight (kg per 667 m^2^)	Total Yield (kg per 667 m^2^)
Ch 0.0	23.65 ± 1.09 ^e^	78.89 ± 2.02 ^e^	102.54 ± 3.73 ^e^
Ch 0.1	27.56 ± 1.69 ^d^	85.69 ± 2.07 ^d^	113.25 ± 2.99 ^d^
Ch 0.5	36.68 ± 0.73 ^c^	117.60 ± 1.92 ^c^	154.28 ± 5.28 ^c^
Ch 1.0	46.86 ± 0.73 ^b^	158.52 ± 1.96 ^b^	205.38 ± 4.40 ^b^
NCC	54.64 ± 1.56 ^a^	190.98 ± 1.49 ^a^	245.62 ± 2.94 ^a^

Values show the mean ± SD, and different letters indicate significant differences at 5% (*p* < 0.05) level.

**Table 3 molecules-28-02053-t003:** The basic information of the *P. ternata* herb garden.

Parameters	Amount	Parameters	Amount
Average altitude	1140 m	Available nitrogen	58.19 mg kg^−1^
Average temperature	16.0 °C	Available phosphorus	4.35 mg kg^−1^
Annual sunshine	1188.7 h	Available potassium	27.04 mg kg^−1^
Annual rainfall	1335.6 mm	Exchangeable calcium	18.66 cmol kg^−1^
pH	6.46	Exchangeable magnesium	311.32 mg kg^−1^
Organic matter	14.51 g kg^−1^	Available zinc	0.65 mg kg^−1^
Total nitrogen	1.42 g kg^−1^	Available iron	6.67 mg kg^−1^
Total phosphorus	1.65 g kg^−1^	Available manganese	15.18 mg kg^−1^
Total potassium	1.13 g kg^−1^	Available boron	0.13 mg kg^−1^

## Data Availability

The datasets used or analyzed during the current study available from the corresponding author upon reasonable request.

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
