# Peer review of "Chitosan Spraying Enhances the Growth, Photosynthesis, and Resistance of Continuous Pinellia ternata and Promotes Its Yield and Quality"

_molecules, 2023, doi:10.3390/molecules28052053_

Round 1

Reviewer 1 Report

I thank MDPI for believing that we can contribute to the merits analysis and publication of the manuscript.

The manuscript is very well written and the justification for carrying it out is very well presented.

Even though I'm not fluent in English, it was possible to detect some language errors. They are highlighted in the manuscript.

Some questions were asked in the manuscript, which I quote below:

1) Keywords have already been mentioned in the title.

2) Was a surfactant used together chitosan?

3) Were the treatments applied twice?

4) We are working with statistic analyses. Therefore, I understand uncorrect to say "slightly lower" (line 191) and after in Discussion item (line 284) you claim about this again. Please review these sentences.

Author Response

Comment 1: I thank MDPI for believing that we can contribute to the merits analysis and publication of the manuscript.

The manuscript is very well written and the justification for carrying it out is very well presented.

Response: We sincerely thank the reviewer for the careful reviews and the positive remarks on our work! The reviewers' comments are all valuable and have been very helpful for revising and improving our manuscript, as well as the important guiding significance to our future researches. Continuous cropping obstacle has become the key factor that seriously restricts the growth, quality, yield and development of Pinellia ternata. This finding highlights that chitosan can be proposed as an alternative and practicable mitigator for alleviating the continuous cropping obstacle of P. ternata. We have studied carefully the reviewers' comments and have made abundant corrections which we hope meet with approval. Thank you most sincerely!

Comment 2: Even though I'm not fluent in English, it was possible to detect some language errors. They are highlighted in the manuscript.

Response: Thanks very much for the reviewer's careful reviews on the manuscript! We did our best to check the English language, and it has been polished by Dr. Kashif Ali Solangi which was marked in blue in the revised manuscript. Thank you most sincerely!

Comment 3: 1) Keywords have already been mentioned in the title.

Response: Special thanks to you for your careful reviews and good comments! The corresponding Keywords have been revised as "Continuous cropping obstacle; alleviation; growth; resistance; quality." which was marked in blue in the revised manuscript. Thank you most sincerely! (See line 26)

Comment 4: 2) Was a surfactant used together chitosan?

Response: Thanks very much for your careful reviews and good comments to our manuscript! This comment is very constructive. In fact, the deacetylation of chitosan used in this study is higher than 90%, and its water solubility is very large. The surfactant mentioned by reviewer may enhance the application effect of chitosan, and we will increase the research in this aspect in future research. The corresponding sentence has been revised as "In conclusion, 0.5~1.0% chitosan spraying could effectively improve the leaf area and plant height of the continuous cropping P. ternata and decline its inverted seedling rate. Meanwhile, 0.5~1.0% chitosan spraying notably enhanced the photosynthesis and resistance of the continuous cropping P. ternata, and effectively improved its yield and quality. This study emphasizes that chitosan can be proposed as an enhancer or mitigator for promoting the growth, yield, and quality of the continuous cropping P. ternata and alleviating its continuous cropping obstacle. However, 0.5~1.0% chitosan could not completely alleviate the cropping obstacle of P. ternata. Thus, the molecular biological mechanism of chitosan for alleviating continuous cropping obstacle of P. ternata and the screening of its synergistic surfactant or adjuvant need to be studied in the future." which was marked in blue in the revised manuscript. Thank you most sincerely! (See lines 324-333)

Comment 5: 3) Were the treatments applied twice?

Response: Thanks very much for the reviewer's careful reviews on our manuscript! The treatment has been carried out twice. P. ternata plant was sprayed by chitosan dilution liquid using an electrostatic atomizer (Qiming Machinery Co. Ltd., Taizhou, Zhejiang, China) on April 15 and April 30, respectively. The dosage of diluent was 60 L per 667 m2. Thank you most sincerely! (See lines 110-113)

Comment 6: 4) We are working with statistic analyses. Therefore, I understand uncorrect to say "slightly lower" (line 191) and after in Discussion item (line 284) you claim about this again. Please review these sentences

Response: Thanks very much for your careful reviews and good advice to our manuscript! This comment is very constructive. "slightly" has been deleted and the corresponding sentence has been revised as "although 0.1~1.0% chitosan spray could not significantly (p < 0.05) increase the chlorophyll of the continuous cropping P. ternate…" which was marked in blue in the revised manuscript. Thank you most sincerely! (See lines 161, 189, 282-283)

As mentioned above, we tried our best to improve the manuscript and made some changes in the revised manuscript. These changes will not influence the content and framework of the manuscript. And here we did not list all the changes but mark them in colour in the body of the revised manuscript.

We appreciate for reviewer's and editor's warm work earnestly, and hope that the correction will meet with approval.

Once again, thank you very much for your comments and suggestions.

Reviewer 2 Report

The study of Chitosan Spraying Enhances the Growth, Photosynthesis, and Resistance of Continuous Cropping Pinellia ternata and Promotes its Yield and Quality which is helpful at agriculture sector. The study will facilitate synthesis of organic fertilizers and help in application of chitosan in agriculture. However, there are some deficiencies which must be addressed.

In abstract the authors are focused on introduction. Here the data collection and main findings should be discussed.

There is no description of methods in the abstract.

Line 66 should be cited with recent studies.

https://doi.org/10.1016/j.bcab.2020.101729 , and DOI:10.1002/aoc.5190

Section 2.4.2 should be cited with relevant study

Line 119 check typos and mistakes.

In introduction discuss sources of chitosan. How chitosan can be prepare easily and economically.

Also types and which type is better for agricultural practices.

Which type was used in this study should be mention in methods.

Conclusion looks like a background of the study. Conclusion must be findings based and proposing gaps ad future recommendations.

Author Response

Comment 1: The study of Chitosan Spraying Enhances the Growth, Photosynthesis, and Resistance of Continuous Cropping Pinellia ternata and Promotes its Yield and Quality which is helpful at agriculture sector. The study will facilitate synthesis of organic fertilizers and help in application of chitosan in agriculture. However, there are some deficiencies which must be addressed.

Response: We sincerely thank the reviewer for the careful reviews and the positive remarks on our work! The reviewers' comments are all valuable and have been very helpful for revising and improving our manuscript, as well as the important guiding significance to our future researches. Continuous cropping obstacle has become the key factor that seriously restricts the growth, quality, yield and development of Pinellia ternata. This finding highlights that chitosan can be proposed as an alternative and practicable mitigator for alleviating the continuous cropping obstacle of P. ternata. We have studied carefully the reviewers' comments and have made abundant corrections which we hope meet with approval. Thank you most sincerely!

Comment 2: In abstract the authors are focused on introduction. Here the data collection and main findings should be discussed.

Response: Thank you very much for your careful reviews and good advice on our manuscript! Same introduction in abstract has been deleted and revised which was marked in blue in the revised manuscript. Thank you most sincerely! (See lines 12-13)

Comment 3: There is no description of methods in the abstract.

Response: Thanks very much for the reviewer's careful reviews and constructive comments! The corresponding method has been added and revised as "In this study, the effects of chitosan on the growth, photosynthesis, resistance, yield, and quality of the continuous cropping P. ternata were investigated by twice field spraying methods." which was marked in blue in the revised manuscript. Thank you most sincerely! (See lines 13-15)

Comment 4: Line 66 should be cited with recent studies.

https://doi.org/10.1016/j.bcab.2020.101729, and DOI:10.1002/aoc.5190

Response: Special thanks to you for your good suggestion! The corresponding literature has been added which was marked in blue in the revised manuscript. Thank you most sincerely! (See lines 65, 405-410)

Comment 5: Section 2.4.2 should be cited with relevant study

Response: Thanks very much for the reviewer's careful reviews and good comments! The corresponding literature has been added as "….at 8:00~10:00 a.m. [27-28]." which was marked in blue in the revised manuscript. Thank you most sincerely! (See line 131)

Comment 6: Line 119 check typos and mistakes.

Response: Thanks very much for the reviewer's hard work and nice comment! The corresponding statement has been revised "The stem diameter was measured by a vernier caliper …" which was marked in blue in the revised manuscript. Thank you most sincerely! (See line 118)

Comment 7: In introduction discuss sources of chitosan. How chitosan can be prepare easily and economically.

Response: Special thanks to you for your hard work and good comment! In fact, commercial chitosan was used in the study, and its deacetylation is higher than 90% so that its water solubility is very large. Thus, we humbly believe that it is very convenient to use in the field. The corresponding statement has been revised "Chitosan, a natural non-toxic N-deacetylation product of chitin to humans or other organisms…" which was marked in blue in the revised manuscript. We sincerely hope to get your understanding and recognition. Thank you most sincerely! (See lines 61-62)

Comment 8: Also types and which type is better for agricultural practices.

Response: Thanks very much for the reviewer's good comment! In fact, commercial chitosan was used in the study, and its deacetylation is higher than 90% so that its water solubility is very large. Thus, we humbly believe that it is very convenient to use in the field. The corresponding statement has been revised as "Water-soluble chitosan (deacetylation ≥90.00%) was produced by Mingrui Bioengineering Co. Ltd. (Zhenzhou, Henan, China)." which was marked in blue in the revised manuscript. We sincerely hope to get your understanding and recognition. Thank you most sincerely! (See lines 86-87)

Comment 9: Which type was used in this study should be mention in methods.

Response: We sincerely thank the reviewer for the careful reviews! The corresponding statement has been revised as "Water-soluble chitosan (deacetylation ≥90.00%) was produced by Mingrui Bioengineering Co. Ltd. (Zhenzhou, Henan, China)." which was marked in blue in the revised manuscript. Thank you most sincerely! (See lines 86-87)

Comment 10: Conclusion looks like a background of the study. Conclusion must be findings based and proposing gaps ad future recommendations.

Response: Thanks very much for the reviewer's good comment! The corresponding sentence has been revised as "In conclusion, 0.5~1.0% chitosan spraying could effectively improve the leaf area and plant height of the continuous cropping P. ternata and decline its inverted seedling rate. Meanwhile, 0.5~1.0% chitosan spraying notably enhanced the photosynthesis and resistance of the continuous cropping P. ternata, and effectively improved its yield and quality. This study emphasizes that chitosan can be proposed as an enhancer or mitigator for promoting the growth, yield, and quality of the continuous cropping P. ternata and alleviating its continuous cropping obstacle. However, 0.5~1.0% chitosan could not completely alleviate the cropping obstacle of P. ternata. Thus, the molecular biological mechanism of chitosan for alleviating continuous cropping obstacle of P. ternata and the screening of its synergistic surfactant or adjuvant need to be studied in the future." which was marked in blue in the revised manuscript. Thank you most sincerely! (See lines 324-333)

As mentioned above, we tried our best to improve the manuscript and made some changes in the revised manuscript. These changes will not influence the content and framework of the manuscript. And here we did not list all the changes but mark them in colour in the body of the revised manuscript.

We appreciate for reviewers and editors' warm work earnestly, and hope that the correction will meet with approval.

Once again, thank you very much for your comments and suggestions.

Reviewer 3 Report

This manuscript by Chen et al seeks to evaluate the ability of chitosan on the growth and certain physiological properties of Pinellia ternata grown for several seasons in a field setting.  The work has seemingly beeen carefuly done and the effects are clear.  However, I feel the English is in need of some improvement before publication.  Generally, the message is clear, but some improvement in the grammar throughout would help.  If you have a native English speaking colleague who would be willing to help, I would suggest that.

1) It is not clear to me that "inverted seedling stage" means.  I tried googling it too, but I couldn't find information on this growth stage.

2) At the end of the introduction, I think it would be beneficial to outline what measurements you will make, and the rationale for choosing each measurement.  I don't really know why you chose to measure "resistance parameters" or why you selected these specific parameters, rather than some other set.  

3) throughout the results, there are many mis-spellings (e.g. "sparying" (spraying), "tredted" (treated), "yeild" (yield)).  Please carefully correct these types of errors.

4) I suggest that you use abbreviations to aid clarity -- for example "NCC" rather than no continuous cropping, or "Ch0.1" etc.

5) I think some consideration of the mechanism of action of chitosan should be discussed in the Discussion.  There are many possible reasons that chitosan may have this effect (e.g. direct nutritive effect, soil pH moderation, changing microbiome, etc), and the positive effects of chitosan may be replicable by another method - provided we know what the real issue is.

Author Response

Comment 1: This manuscript by Chen et al seeks to evaluate the ability of chitosan on the growth and certain physiological properties of Pinellia ternata grown for several seasons in a field setting.  The work has seemingly beeen carefuly done and the effects are clear.  However, I feel the English is in need of some improvement before publication.  Generally, the message is clear, but some improvement in the grammar throughout would help.  If you have a native English speaking colleague who would be willing to help, I would suggest that.

Response: We sincerely thank the reviewer for the careful reviews and the positive remarks on our work! The reviewers' comments are all valuable and have been very helpful for revising and improving our manuscript, as well as the important guiding significance to our future researches. Continuous cropping obstacle has become the key factor that seriously restricts the growth, quality, yield and development of Pinellia ternata. This finding highlights that chitosan can be proposed as an alternative and practicable mitigator for alleviating the continuous cropping obstacle of P. ternata. We have studied carefully the reviewers' comments and have made abundant corrections which we hope meet with approval. We also did our best to check the English language, and it has been polished by Dr. Kashif Ali Solangi which was marked in blue in the revised manuscript. Thank you most sincerely!

Comment 2: 1) It is not clear to me that "inverted seedling stage" means.  I tried googling it too, but I couldn't find information on this growth stage.

Response: Thanks very much for the reviewer's careful reviews on the manuscript! The inverted seedling is a physiological phenomenon that often occurs during the growth of P. ternata, which is a dormancy to resist adverse environment. During the growth of P. ternata, when the environmental conditions such as temperature, humidity, light, allelochemicals, etc. change greatly, its aboveground stems and leaves will gradually wither, and the inverted shape (inverted seedling) makes its underground stems to survive the adverse environment. The corresponding statement have been added and revised as "The inverted seedling is a physiological phenomenon that often occurs during the growth of P. ternata, which is a dormancy to resist adverse environment. During the growth of P. ternata, when the environmental conditions such as temperature, humidity, light, allelochemicals, etc. change greatly, its aboveground stems and leaves will gradually wither, and the inverted shape (inverted seedling) makes its underground stems to survive the adverse environment. " and " These findings here emphasize that 0.5~1.0% chitosan spary could effectively decrease the inverted seedling rate of P. ternata, and alleviate its cropping obstacle but could not be completely alleviated, which might be related to chitosan application to enhance the stress resistance of P. ternata." which were marked in blue in the revised manuscript. Thank you most sincerely! (See lines 180-185, 190-194)

Comment 3: 2) At the end of the introduction, I think it would be beneficial to outline what measurements you will make, and the rationale for choosing each measurement.  I don't really know why you chose to measure "resistance parameters" or why you selected these specific parameters, rather than some other set.

Response: Special thanks to you for your careful reviews and good comments! This comment is very constructive. The corresponding statement has been revised as "In this work, the influences of chitosan on the growth (leaf area, plant height, stem diameter, and inverted seedling rate) and photosynthesis (chlorophyll, photosynthetic rate (Pn), intercellular carbon dioxide concentration (Ci), stomatal conductance (Gs), and transpiration rate (Tr)) of the continuous cropping P. ternata was evaluated. Simultaneously, the stress resistance capacity of plants strongly affects their growth, soluble sugar, proline (Pro), malonaldehyde (MDA), superoxide dismutase (SOD), peroxidase (POD), and catalase (CAT) are closely connected to stress resistance. Thus, the influences of chitosan on the abovementioned resistance parameters of continuous cropping P. ternata was assessed. Finally…" which was marked in blue in the revised manuscript. Thank you most sincerely! (See lines 79-87)

Comment 4: 3) throughout the results, there are many mis-spellings (e.g. "sparying" (spraying), "tredted" (treated), "yeild" (yield)).  Please carefully correct these types of errors.

Response: Special thanks to you for your careful reviews and good comments! We did our best to check the English language, and it has been polished by Dr. Kashif Ali Solangi which was marked in blue in the revised manuscript. Thank you most sincerely!

Comment 5: 4) I suggest that you use abbreviations to aid clarity -- for example "NCC" rather than no continuous cropping, or "Ch0.1" etc.

Response: Thanks very much for your careful reviews and good comments to our manuscript! This comment is very constructive. The corresponding abbreviations has been revised and stated which were marked in blue in the revised manuscript. We sincerely hope to get your understanding and recognition! Thank you most sincerely! (See lines 111-115)

Comment 6: 5) I think some consideration of the mechanism of action of chitosan should be discussed in the Discussion. There are many possible reasons that chitosan may have this effect (e.g. direct nutritive effect, soil pH moderation, changing microbiome, etc), and the positive effects of chitosan may be replicable by another method - provided we know what the real issue is.

Response: Thanks very much for your careful reviews and good comments to our manuscript! This comment is very constructive. The corresponding statement has been revised as " Overall, the possible mechanism of chitosan mediated the alleviation of continuous cropping obstacle of P. ternata was as follows: chitosan might be used as a plant growth promoter or biological fertilizer to enhance the growth and photosynthesis of continuous cropping P. ternata, and might also be used as an inducer to improve its stress resistance and reduce its inverted seedling rate, as well as might also be used as a biological fungicide to prevent its disease occurrence. These possible positive effects together enhanced the growth, yield, and quality of continuous cropping P. ternata and alleviated its continuous cropping obstacles. In the future, the molecular biological mechanism of chitosan for alleviating continuous cropping obstacle of P. ternata needs to be studied to better understand the alleviation mechanism of chitosan." which was marked in blue in the revised manuscript. Thank you most sincerely! (See lines 345-355)

As mentioned above, we tried our best to improve the manuscript and made some changes in the revised manuscript. These changes will not influence the content and framework of the manuscript. And here we did not list all the changes but mark them in colour in the body of the revised manuscript.

We appreciate for reviewer's and editor's warm work earnestly, and hope that the correction will meet with approval.

Once again, thank you very much for your comments and suggestions.